# The LipoDerm Method for Regeneration and Reconstruction in Plastic Surgery: A Technical Experimental Ex Vivo Note

**DOI:** 10.3390/medsci11010016

**Published:** 2023-02-03

**Authors:** Ziyad Alharbi, Sarah Qari, Maryam Bader, Sherif Khamis, Faris Almarzouqi, Michael Vogt, Christian Opländer

**Affiliations:** 1Plastic Surgery and Burn Unit, Dr. Solaiman Fakeeh Hospital, Jeddah 21461, Saudi Arabia; 2Clinical Sciences Department, Fakeeh College for Medical Sciences, Jeddah 21461, Saudi Arabia; 3Head of Plastic Surgery Department, International Medical Center, Jeddah 21423, Saudi Arabia; 4Interdisciplinary Center for Clinical Research, RWTH Aachen, 52074 Aachen, Germany; 5Institute for Research in Operative Medicine (IFOM), University Witten/Herdecke, 51109 Cologne, Germany

**Keywords:** dermal scaffolds, fat grafting, adipose-derived stem cells, lipoaspirate, nanofat, tissue engineering, soft tissue defects

## Abstract

The combination of adipose-derived stem cells (ASCs) and dermal scaffolds has been shown to be an approach with high potential in soft tissue reconstruction. The addition of dermal templates to skin grafts can increase graft survival through angiogenesis, improve regeneration and healing time, and enhance the overall appearance. However, it remains unknown whether the addition of nanofat-containing ASCs to this construct could effectively facilitate the creation of a multi-layer biological regenerative graft, which could possibly be used for soft tissue reconstruction in the future in a single operation. Initially, microfat was harvested using Coleman’s technique, then isolated through the strict protocol using Tonnard’s technique. Finally, centrifugation, emulsification, and filtration were conducted to seed the filtered nanofat-containing ASCs onto Matriderm for sterile ex vivo cellular enrichment. After seeding, a resazurin-based reagent was added, and the construct was visualized using two-photon microscopy. Within 1 h of incubation, viable ASCs were detected and attached to the top layer of the scaffold. This experimental ex vivo note opens more dimensions and horizons towards the combination of ASCs and collagen–elastin matrices (i.e., dermal scaffolds) as an effective approach in soft tissue regeneration. The proposed multi-layered structure containing nanofat and dermal template (Lipoderm) may be used, in the future, as a biological regenerative graft for wound defect reconstruction and regeneration in a single operation and can also be combined with skin grafts. Such protocols may optimize the skin graft results by creating a multi-layer soft tissue reconstruction template, leading to more optimal regeneration and aesthetic outcomes.

## 1. Introduction

Dermal substitutes composed of collagen–elastin matrices, which mimic the lost dermis, are biocompatible scaffolds commonly used for soft tissue reconstruction and tissue-engineering approaches [1]. As a result, they provide a solution to undesirable functional and aesthetic outcomes that arise from the solitary use of split-thickness skin grafts (STSG), including contractures, decreased skin elasticity, and mismatched skin color. Min et al. have demonstrated that the coverage of full-thickness skin defects by using a combination of Matriderm and STSG in a single operation solved these concerns, improving scar formation, skin elasticity, and skin color [2]. Combining these scaffolds with adipose-derived stem cells (ASCs) can introduce additional benefits in multi-layer soft tissue reconstruction, by providing the missing hypodermis layer in order to resemble the normal skin architecture [1,3].

Common universally used scaffolds include Integra and Matriderm. Both dermal substitutes provide good results in complex wound healing with areas of skin loss due to trauma or chronic inflammation involving different types of wounds. Matriderm is composed of bovine dermal collagen I, III, and V and elastin without cross-linking, whereas Integra consists of bovine tendon collagen type I and shark glycosaminoglycan cross-linked with glutaraldehyde [4]. The reason we favored the use of Matriderm over Integra is the ability to use Matriderm in combination with skin grafts within a single procedure, whereas Integra requires a two-step procedure. Matriderm also has the beneficial addition of elastin, providing more elasticity and mechanical resilience. In addition to providing similar results in wound healing, Matriderm has a lower cost [5]. Moreover, Böttcher-Haberzeth et al. conducted a comparative study between the coverage of full-thickness wound defects by Matriderm and Integra on rats. The Matriderm-based neodermis was found to be thinner but showed a higher cell density than the Integra-based neodermis, thus providing benefits to help with the diffusion of blood and nutrients [6].

ASCs are abundantly present in adipose tissue and have been shown to secrete growth factors—mainly vascular endothelial growth factors (VEGFs), which have an angiogenic and anti-apoptotic effect—as well as being able to proliferate and differentiate into multiple cell lineages [7,8]. This can stimulate wound healing, providing an accelerated healing rate and decreased scar formation [9,10]. This makes them crucial for stem cell-based therapies and validates their evident exploitation in tissue regeneration and reconstruction [3,11].

We have previously shown that lipoaspirates obtained from multi-port microcannulas 0.7 mm in diameter producing microfat (as per Coleman’s technique) produced a greater ACS yield, as well as increased migration and adherence rate to collagen–elastin matrices, in comparison to macrocannulas producing macrofat. This study indicated that the liposuction method with different cannula sizes affects graft survival and improves the cosmetic aspect of scars [12,13].

Based on the technique of adipocyte harvesting and its manipulation, there are many types of fat grafts. Nanofat is produced by the filtration of microfat, harvested through liposuction using Coleman’s technique. Nanofat is a new technique that has recently gained popularity in cosmetic surgery. Studies have shown that nanofat contains no viable adipocytes but is more abundant in stromal vascular fraction cells and adipose-derived stem cells [14], thereby enhancing skin quality and promoting tissue regeneration. It is also liquid, in comparison to other types of fat, due to the absence of adipocytes, which means that it lacks a volumizing effect [15]. In contrast, microfat contains many viable adipocytes, offering more soft tissue structural support, and is mainly utilized for its filling effect [13,16,17,18].

In a previous study, collagen–elastin matrices were incubated with nanofat-containing ASCs to assess cellular enrichment at specific periods of time (1, 3, and 24 h). As time progressed, the cellular enrichment was amplified and, histologically, the seeded ACSs penetrated deeper into the matrices, but cells had adhered to the matrix even at 1 h. This verifies the possibility of ASC transfer onto collagen–elastin matrices in a single operation for multi-layer soft tissue reconstruction. These results also indicate that dermal scaffolds—specifically, collagen–elastin matrices—are cytocompatible and are suitable for use with ASCs immediately after their isolation [1].

After grafting, chemical antiseptics and saline are required to rinse the wounds, as well as in dressings following the grafting procedure, which is considered a possible risk factor that may affect the viability of ASCs. An in vitro trial has concluded that ASCs had a minimal negative effect in treatment with mafenide acetate and saline, which may be regarded as a feasible antiseptic due to its low ASC toxicity compared with other antiseptics (which showed moderate to severe toxicity) [19].

Adipose tissue engineering offers a novel solution for soft tissue defects. Through the development of bioactive tissue constructs, fatty tissue can be regenerated, in terms of both structure and function, as a soft tissue replacement. Volume loss, color discrepancy, and donor site morbidity over time are considered significant pitfalls of current tissue flap methods [20]. Our model consists of ASCs acquired by liposuction using a multi-port microcannula, followed by refining of the obtained microfat into nanofat. Finally, the nanofat is applied and absorbed by the Matriderm, thus generating the Lipoderm.

The autologous use of ASCs on top of collagen–elastin matrices has been previously considered, showing good outcomes in multiple published articles [21]. The use of Lipoderm in multi-layer soft tissue reconstruction led to increased graft survival through improved angiogenesis, anti-inflammatory properties, accumulation, and promotion of collagen, along with anti-scarring therapeutic benefits [22], thus improving the regeneration capacity and healing time, as well as enhancing the overall appearance of the skin graft.

## 2. Materials and Methods

### 2.1. Centrifugation of Lipoaspirate Obtained by Liposuction

Fat harvesting was achieved using an st’RIM cannula (Thiebaud Biomedical Devices, Margencel, France), as established by Guy Megalon, through tumescent liposuction. The blunt tip cannula was 2 mm in diameter and contained four 600 μm gauge orifices. The harvested microfat was subsequently centrifuged in a Sigma 2–16 K centrifuge (Osterode am Harz, Germany) for 3 min at 3000 rpm. This produced the purified lipoaspirates, which were then immediately used in the experiments (Figure 1).

### 2.2. Extraction of SVF/ASCs from Lipoaspirate

The obtained purified lipoaspirates were transported into a sterile tube, where normal saline was added to remove cell debris and blood. We conducted centrifugation for a second time, for 10 min at 300× *g*. A 45 min digestion of the extracellular matrix at 37 °C was performed with 0.075% collagenase I (Biochrom, Berlin, Germany), and the samples were subsequently filtered using a 250 μm filter (Neolab, Heidelberg, Germany). The pellet was then re-suspended in 30 mL of a NaCl solution and centrifuged for another 10 min at 300× *g* in order to acquire the stromal vascular fraction containing ASCs. The pellet was then re-suspended for a final time in DMEM/F12 and strengthened using 100 U/mL of penicillin and 100 μg of streptomycin. The isolated cells were then directly transplanted from the pellet onto the collagen–elastin matrix as shown in (Figure 2).

### 2.3. Assessment of Cellular Adherence of ASCs onto Collagen–Elastin Matrices

A collagen–elastin matrix derived from bovine skin, containing type I, III, and V collagen (Matriderm^®^ sheet; MedSkin Solutions Dr. Suwelack AG, Billerbeck, Germany), was cut into 1 mm thick circular pieces using a 0.8 cm diameter punch biopsy. The collagen–elastin matrix pieces were then placed into 48-well culture plates at a density of 50,000 cells per well, to which the isolated ASCs were added, then incubated at 37 °C with 5% CO_2_. After an incubation period of 1, 3, or 24 h, the matrix and ASCs complex were separated. Normal saline (0.9% NaCl) was used to wash the pieces carefully, which were then shifted to a culture plate. A 270 μL volume of DMEM/F12 strengthened using 100 U/mL of penicillin and 100 μg of streptomycin was then added into each well, in addition to 30 μL of the alamarBlue^®^ resazurin reagent (AbD Serotec, Oxford, UK). Using a fluorescence spectrometer, the assay described below was conducted to assess cellular metabolism. A repeat of the medium/alamarBlue^®^ mix was carried out at 37 °C and 5% CO_2_ and, after 2 h, was carefully removed from the well. Finally, the samples were measured using a Fluostar Optima fluorescence spectrometer (BMG Labtech, Offenburg, Germany) at room temperature, with an excitation wavelength of 540 nm and an emission wavelength of 590 nm. The matrix itself was not measured, in order to prevent the influence of the matrix on fluorescence. Negative control was obtained by adding alamarBlue^®^ reagent without cells.

## 3. Results

### Utilizing Two Photon Microscopy to Analyze Cellular Distribution following Incubation

To visualize the 3D structure and organization of the ASCs within the matrix, two-photon microscopy using a FV1000MPE microscope (Olympus Corp., Tokyo, Japan) connected to a pulsed Ti–Sapphire laser (MaiTai DeepSee, SpectraPhysics, Santa Clara, CA, USA) was carried out. Furthermore, fluorescein diacetate (FDA) staining was conducted to show the viable isolated cells, while Hoechst 33,342 staining was conducted to visualize the nuclei. Finally, the matrix was visualized using the non-linear optical effect of second harmonic generation (SHG). Hoechst was excited at 730 nm and detected at 418–468 nm. A series of subsequent 1024 × 1024 pixel xy-frames were then obtained in 1 mm z-steps for structural 3D reconstruction using the Imaris Software (Bitplane, Zurich, Switzerland); see Figure 3.

## 4. Discussion

The regeneration and reconstruction of soft tissue defects can be challenging, as current methods are commonly associated with issues such as volume loss, color discrepancy, and a possibility of donor site morbidity over time [20].

Adipose tissue engineering and multi-layer soft tissue reconstruction offer novel solutions to these problems. Numerous articles in the literature have recently proved that dermal scaffolds have an advantage over skin grafts in decreasing the infection rate and healing time. Additionally, the increase in elasticity helps the skin to better recover its initial shape [23,24].

The addition of ASCs to such a construct provides further benefits, due to their ability to accelerate wound healing through the reactivation of dermal fibroblasts and keratinocytes, angiogenesis, and differentiation [8,10,25]. Nie et al. tested this theory through an experiment on 36 normal and 36 diabetic rats, where ACSs were incubated in vitro. Their results showed that ASCs promoted wound healing and reduced the healing time in both groups through differentiation to multiple cell lineages (as observed by immunofluorescence), secretion of angiogenic growth factors (mainly VEGF), along with increased epithelization and granulation, in comparison with control wounds [26].

We have previously shown that smaller cannulas resulted in higher viability of ASCs than bigger cannulas [12]. Moreover, in previous publications, the successful ex vivo integration of ASCs into collagen–elastin matrices after a 3 h incubation period—and, more interestingly, even at 1 h—has been proven. This time was sufficient for integration, confirming that it is possible to manufacture a construct ex vivo. Therefore, we hypothesized and proposed the formation concept of the so-called Lipoderm (Figure 3) [1].

After successful integration, proliferation followed by the differentiation process of ASCs into the adipogenic lineage took place, as shown in Figure 3. These results, analyzed by the 2-photon microscope, indicated that the nanofat seeded onto the collagen–elastin matrix showed a high yield of proliferated and differentiated ASCs (ex vivo). We anticipate that such a construct can not only cover the skin defect, but also aids in regenerating the lost subcutaneous fat, providing solutions for the downsides of current soft tissue reconstruction, such as volume loss and donor site morbidity. The ability of ASCs to regenerate the adipogenic structure, combined with the autologous fat grafting technique, leads to an increase in angiogenesis through the growth factors present in the lipoaspirate, decreasing color discrepancy and donor site morbidity over time and preventing volume loss, thus resulting in better aesthetic and functional results.

The solitary use of dermal scaffolds along with skin grafts has been shown to be effective in covering areas of skin loss. It has been speculated that the increase in the distance between the wound bed and the graft due to dermal substitutes may hinder vascular ingrowth and limit the diffusion of nutrients to the graft. A previous study compared dermal substitutes of different thicknesses to the use of STSG alone in a single-stage procedure, and it was found that the thinner dermal scaffolds performed similarly to the solitary use of split-thickness skin grafts (STSG), while the thicker substitutes presented worse graft take [27].

The benefits obtained through the addition of dermal scaffolds along with skin grafts include an improvement in scar formation, skin elasticity, skin color, and prevention of the formation of contractures, in comparison with the solitary use of STSG. Dermal scaffolds also provide a solution when there are exposed tendons, bone, and other structures [2,28,29].

Cervelli et al. compared the effectiveness of Matriderm combined with skin grafts to the isolated use of autologous skin grafts in post-traumatic wounds, on the basis of Manchester Scar Scale (MSS) and patient satisfaction. They found the healing and re-epithelization time to be superior with the use of Matriderm within 15 days, and differences were still observed up to 3 months post-operation. It was also found that the use of Matriderm reduced wound contractures and scar formation, as well as providing better skin elasticity, architecture, and overall appearance [28].

The SVF/ASC obtained from lipoaspirates contain multi-potent stem cells, progenitor cells, and growth factors—most importantly, vascular endothelial growth factors (VEGFs)—which have previously been shown to promote the rapid formation and ingrowth of microvascular structure on dermal scaffolds [21,30,31,32]. Später et al. have demonstrated that ASCs were superior, when compared with SVF, in terms of neovascularization and collagen deposition [33]. This would suggest that single-stage procedures may be viable with the use of thinner dermal substitutes, even more so if the dermal substitutes were infiltrated with ASCs, thus promoting angiogenesis and graft take [27].

Clinically, the solitary use of dermal scaffolds along with skin grafts has been verified, in terms of its efficacy in covering moderate to large areas of skin loss with limited regenerative healing capabilities, in terms of skin color, softness, and thickness. In reconstruction and regeneration, repairing all three layers of the epidermis, dermis, and hypodermis to enhance skin function and beautifying the final result are always the most important pursuits.

Our model was completely carried out ex vivo and the results, along with existing supportive evidence, indicated improved results when combining ASCs with collagen–elastin matrices. This attests to our proposal to use the Lipoderm in vivo to attain an enhanced level of multi-layer healing, in order to refine the overall appearance and improve skin function.

Normally, we apply water (Sterile NS) onto Matriderm in order to cause it to attach to the ground of the wound; this is the established protocol for surgeons worldwide. Lipoconcentrate and nanofat products are very fine and liquid in nature; thus, they can be transferred to the sub-dermal skin substitutes, instead of NS, following which the product can be taken by hand directly to be inserted into the ground of the wound. This application would be made easier if the Matriderm structure is first transplanted onto the wound site, after which the ACSs can be distributed in situ to the wound site; furthermore, in order to avoid the breakdown of the tissue, definitive closure of the last layer of skin can be obtained by use of STSG as an outer coverage for the wound.

One important fact is that many surgeons would like to proceed with a single-stage reconstructive procedure, instead of a two-stage procedure. For example (but not limited to), Integra Dermal Regenerative Substitutes are still one of the common alternatives; however, their use requires a two-stage procedure, rather than a one-stage procedure. It can be applied to the wound, upon which a silicone cover is placed for several weeks, until granulation and regeneration of the wound takes place, following which a skin graft can be applied. On the other hand, some regenerative templates, such as Matriderm, include collagen and elastin and have the ability to be applied in a one-stage procedure, in addition to the skin grafting [4,5,6]. Finally, the addition of adipose-derived stem and stromal cells would be favorable for many reasons, including the formation of fat tissue (either as volume or as regeneration), in addition to the promising healing ability of these cells regarding the wound and the scars.

At present, fat grafting is evolving rapidly, with many techniques including the use of microfat, lipoconcentrate, and nanofat [34]. The main difference between lipoconcentrate and microfat, in comparison to nanofat, is the ability to enhance volume in the body or face rather than a focus on regeneration, which is more enhanced in the latter. When the skin is then transplanted over the lipoderm construct, we can later enhance the general appearance of the skin; for example, using re-touch techniques, which may lead to significant improvements in terms of color or quality, PRP micro-needling, and/or CO_2_ fractional laser treatment, as we have previously studied [34].

Lipoderm treatment may provide a promising technique for complex wound regeneration, especially for wounds with exposed vital structures, where complete fat regeneration is preferred with dermal and sub-dermal repair, such that multi-layer soft tissue reconstruction could be obtained in one day in the clinical setting, through established and safe techniques without the need for further processing in labs. This is expected to minimize the cost and time for both patients and healthcare systems.

In the future, such constructs may open up a variety of horizons for single-stage repair procedures in a wide range of surgical domains and for a variety of defects, especially complex defects that would usually require significant procedures such as regional or free flaps. Reconstruction of such defects through this approach would provide the patient the benefits of a shorter procedure time, as well as less complications and bleeding risks, due to the use of biomaterials and autologous products. Furthermore, the procedure could be performed as a day-case surgery, and the patient may be discharged in the same day.

SVF cells play one of the most important roles clinically, especially when tissue engineering protocols are to be used for patients. Through this ex vivo experimental note, we shed light on the efficiency and safety regarding the possible use of such products. Nevertheless, we should clearly state that further analytical research should also be initiated, in order to identify the best quantity of cells needed (e.g., per square meter) before transplantation.

## 5. Conclusions

Clinically, the efficacy of the solitary use of dermal scaffolds along with skin grafts has been verified, in terms of covering moderate to large areas of skin loss, while presenting limited regenerative healing capabilities, in terms of skin color, softness, and thickness. In reconstruction and regeneration, repairing all three layers of the epidermis, dermis, and hypodermis to enhance their functions and to beautify the final result are always the most important pursuits.

Our model was completely carried out ex vivo and its result, along with existing supportive evidence, demonstrated improved results through the combination of ASCs with collagen–elastin matrices. This validates our proposal to use Lipoderm in vivo, in order to attain an enhanced level of multi-layer healing to refine the overall appearance and improve skin functions.

Lipoderm appears to be a promising technique for complex wound regeneration, especially for wounds with exposed vital structures where complete fat regeneration is preferred with dermal and sub-dermal repair, such that multi-layer soft tissue reconstruction could be carried out in one day in the clinical setting using established and safe techniques, without the need of further processing in labs. This is expected to minimize the cost and time for both patients and healthcare systems.

## Figures and Tables

**Figure 1 medsci-11-00016-f001:**
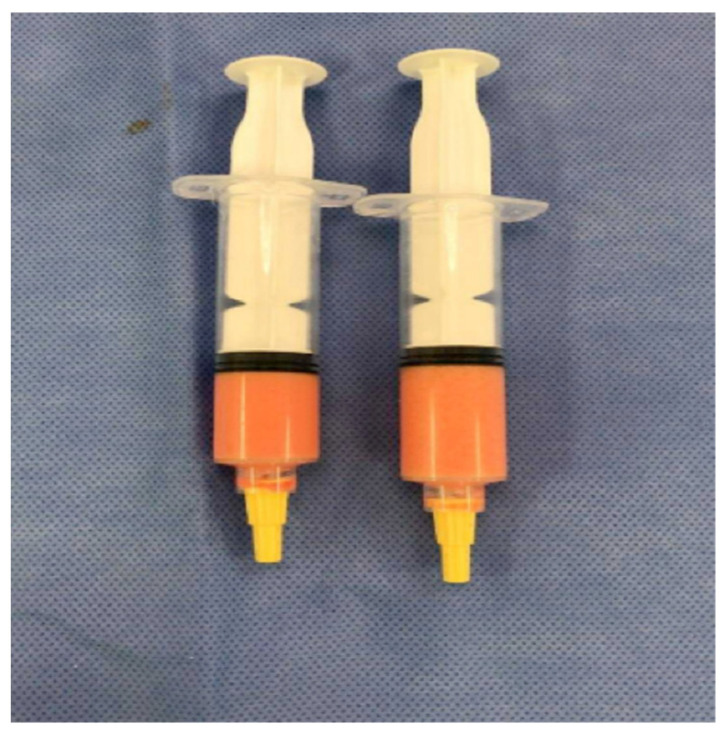
Purified lipoaspirate transferred into a sterile tube following removal of cell debris and blood.

**Figure 2 medsci-11-00016-f002:**
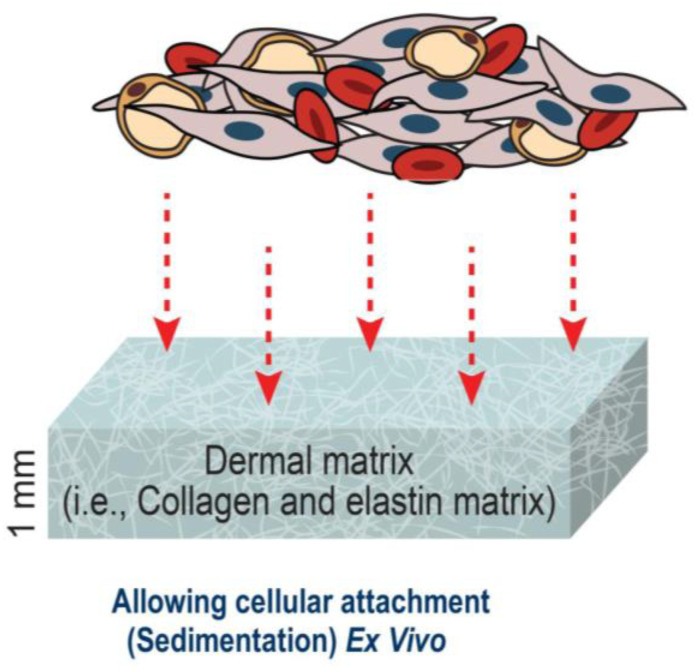
Application of ASCs onto collagen–elastin matrices.

**Figure 3 medsci-11-00016-f003:**
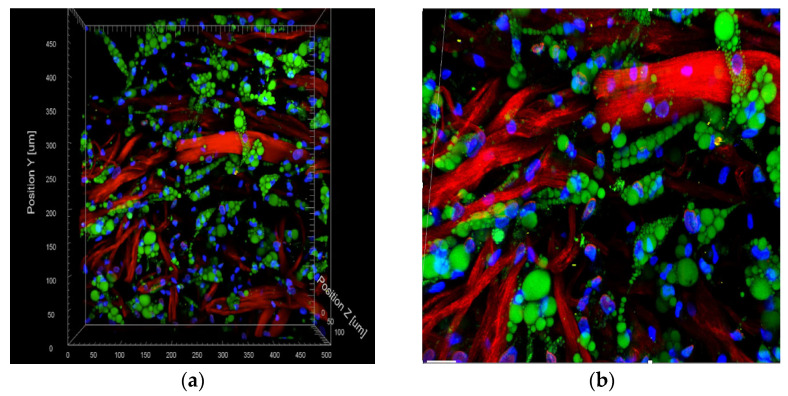
(**a**,**b**): Two-photon microscopy image of a collagen–elastin matrix incubated with freshly isolated ASCs after 1 h. The cytoplasm of viable cells (represented in green) was stained using fluorescein diacetate. The cell nuclei (represented in blue) were stained using Hoechst 33,342. Finally, the red color represents the collagen–elastin matrix (100 μm scale).

## Data Availability

Not applicable.

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
