# Peer review of "The LipoDerm Method for Regeneration and Reconstruction in Plastic Surgery: A Technical Experimental Ex Vivo Note"

_medsci, 2023, doi:10.3390/medsci11010016_

Round 1

Reviewer 1 Report (Previous Reviewer 4)

Congratulations for your novel approach to difficult wounds; I totally agree with you that the autologous regenerative therapy is opening a new era in vulnology as in other medical fields; further research should be initiated .

Author Response

Thank you, very much dear colleague, for such a response. We are very happy that we could be able to meet your points after our revision to the manuscript.

Reviewer 2 Report (Previous Reviewer 3)

All my suggestions for improvement have been implemented

Author Response

Thank you for such a response. We are very happy that we could be able to meet your points after our revision to the manuscript.

Reviewer 3 Report (Previous Reviewer 2)

Authors did not show any additional data that are crucial in judging the value of the manuscript. The volume of the data were even reduced due to deletion of Figure 4, which had once been expected to show the clinical significance of the current technique. The reviewer thinks that the resubmitted manuscript does not meet the standard of the publication in Medical Science. 

Author Response

Thank you for this response which is very important for us. We have deleted the picture as It has made some confusion for other reviewer but we will make sure to modify so we can implement your comment which is very important for us.

Reviewer 4 Report (Previous Reviewer 1)

I can find no significant improvement in the manuscript.

Author Response

Thank you, for such a response. We are making sure to make an extensive format editing for the manuscript including extensive English editing. Furthermore, the raised points will be implanted carefully in the discussion part. We will also make sure to insight the light on available data and the need of further research in this filed.

Round 2

Reviewer 3 Report (Previous Reviewer 2)

Authors did not show any additional data that are crucial in judging the value of the manuscript again in the second revision. Since the data shown in the current manuscript is identical to those in the previous one, the judgement (Rejection) cannot be changed. 

Reviewer 4 Report (Previous Reviewer 1)

no comments

This manuscript is a resubmission of an earlier submission. The following is a list of the peer review reports and author responses from that submission.

Round 1

Reviewer 1 Report

General Comments

This is meant to be a technical note for the use of ASCs combined with a collagen/elastin matrix for plastic surgical applications.

The manuscript is chaotic and poorly written.  The authors are harvesting tissue/cells from human subject apparently without proper Institutional Review Board approval.

Specific Comments

Introduction

P1, l38; Dermal substitutes  There are many different types of dermal substitutes not just collagen elastin matrices.

Materials and Methods

The authors need to describe the subject population and provide evidence for patient consent.  Without this consent the manuscript must be rejected.

The results describe the histology of ASC seeded matrices.  However, the discussion also describes the clinical application of the product.  There is no mention of this in either the Methods or Results section.

Author Response

Point 1: Dermal substitutes  There are many different types of dermal substitutes not just collagen elastin matrices.

Response 1: The dermal substitutes we have used are collagen elastin matrices, we changed the wording in the introduction to state that we were talking about dermal scaffolds composed of collagen elastin matrices

Point 2: The authors need to describe the subject population and provide evidence for patient consent.  Without this consent the manuscript must be rejected.

Response 2: The fat tissue has been taken after conventional liposuction was done for patients. Normally fat after liposuction will be discarded in the OR if lipofilling or fat grafting will not be used for the same patients. For these categories the fat may not be discarded in OR and used for further research in the lab without any further clinical intervention to the patient. For such lab usage ethical approval is attached taken from the Aachen university hospital ethical committee.

Point 3: The results describe the histology of ASC seeded matrices.  However, the discussion also describes the clinical application of the product.  There is no mention of this in either the Methods or Results section.

Response 3: We proposed future clinical application to attain multilayer healing that would improve overall appearance and function. Our model and existing supportive evidence supports this conclusion.

Reviewer 2 Report

Authors claim that only one hour incubation of isolated Nano-Fat Lipoaspirate onto collagen and elastin matrices provide LipoDerm that would contribute to regeneration of multilayer skin healing. However, the presentation lacks sufficient evidence to support the conclusions. Therefore, it is hard to judge the scientific soundness in the current form. Resubmission after solving the following concerns is recommended. 

Major concerns:

1) In Figure 3, there are no data for control conditions. It also lacks data with quantitative analyses. Please provide sufficient data that prove the author's claims.  

2) In Figure 3, the major green signals are derived from the autofluorescence of lipid droplets in mature adipocytes but not from fluorescein diacetate (FDA), which  stains the cytoplasm of viable cells. Actually, it is very hard to detect the FDA-positive ASC, which morphologically resembles fibroblasts. Please provide new data where autofluorescence signals are completely eliminated.

3) Since there are no descriptions regarding Figure 4 in the main text, it is very hard to assess the potential of LipoDerm that authors provided in the current study. Please add descriptions regarding the transplantation procedure along with its approval of the Ethics Committee in "Materials and Methods" and detailed explanations regarding the finding of Figure 4 in "Results". 

Minor concurs:

1) Figure 3 contains no the "grey" materials. Please add arrows that indicate the grey-colored collagen and elastin structure. Also, please add descriptions regarding "red" materials in legend of Figure 3. 

Author Response

Point 1: In Figure 3, there are no data for control conditions. It also lacks data with quantitative analyses. Please provide sufficient data that prove the author's claims.  

Response 1: For figure 3, this experiment is used as a pilot point which definitely needs further analysis with a big cohort of patients. We therefore would like to insight the light, through this publication, about the possible integration of seeded cells with collagen elastin matrices represented by subdermal skin substitutes.

Point 2: In Figure 3, the major green signals are derived from the autofluorescence of lipid droplets in mature adipocytes but not from fluorescein diacetate (FDA), which  stains the cytoplasm of viable cells. Actually, it is very hard to detect the FDA-positive ASC, which morphologically resembles fibroblasts. Please provide new data where autofluorescence signals are completely eliminated.

Response 2: We agree that the green signals represent the mature adipocyte fat droplets and that was one of our intentions to indicate that different differentiation took place after successful segmentation of ASCs and stromal cells on the scaffolds. We therefore would like to state that a successfully differentiated fat tissue would be an area of the target needed for clinicians especially for wound regeneration such as in cases of exposed tendons and bone where complete regeneration of fat tissue is needed.

Point 3: Since there are no descriptions regarding Figure 4 in the main text, it is very hard to assess the potential of LipoDerm that authors provided in the current study. Please add descriptions regarding the transplantation procedure along with its approval of the Ethics Committee in "Materials and Methods" and detailed explanations regarding the finding of Figure 4 in "Results". 

Response 3: This was an ex-vivo experiment so we decided to remove figure 4 as it could be confusing to the reader and focus on the experiment result. We still have not done any clinical trials using this method

Point 4: Figure 3 contains no the "grey" materials. Please add arrows that indicate the grey-colored collagen and elastin structure. Also, please add descriptions regarding "red" materials in legend of Figure 3. 

Response 4: Changes in figure 3 captions were applied.

Reviewer 3 Report

The manuscript is well written and covers an interesting topic. 

I have 3 small suggestions for improvement:

1) Please describe the novelty of your work in a few sentences.

2) I find the approach of transplanting the combination of Matriderm and nanofat very interesting. Because Matriderm alone increases the dissusion distance for split skin even without nanofat, it can be assumed that the diffusion distance with nanofat is even higher. This should be discussed at least briefly.

3) How stable is the product after the application of nanofat? is it still transplantable? please also briefly discuss.

Author Response

Point 1: Please describe the novelty of your work in a few sentences.

Response 1: Regarding the applicability of the proposed matrix in which we have a dermal scaffold in this case collagen elastin matrix which can be enriched with adipose derived stem cells obtained by nanofat or lipoconcentrate technology. This application would be easier if the matriderm structure is first transplanted to the wound site then the ASCs can be disturbuted in citu also on the wound site  in order to avoid the breakdown of tissue. Then definitive closure of the last layer of skin can be obtained by STSG as an outer coverage for the wound. Lipoderm would be a promising technique for complex wound regeneration, especially for wounds with exposed vital structures where complete fat regeneration is preferred with dermal and subdermal repair so that a multilayer soft tissue reconstruction could be obtained one day in the clinical setting with established and safe techniques without the need of further processes in labs. In order to minimize the cost and time for patients and healthcare systems.

Point 2: I find the approach of transplanting the combination of Matriderm and nanofat very interesting. Because Matriderm alone increases the dissusion distance for split skin even without nanofat, it can be assumed that the diffusion distance with nanofat is even higher. This should be discussed at least briefly.

Response 2: It has been added to the article with citations.

Point 3: How stable is the product after the application of nanofat? is it still transplantable? please also briefly discuss.

Response 3: Normally we apply Sterile NS on the matriderm in order to make it attached to the ground of the wound and this is the established protocol for surgeons around the world. Lipoconcentrate and Nanofat products are very fine and liquid in nature so that they can be transferred to the subdermal skin substitutes instead of NS and then the product can be taken by hand directly to be inserted in the ground of the wound.

Reviewer 4 Report

In the regenerative therapy the amount of SVF cells is very important in evaluating the effectiveness of the treatment.  In this study the quantity and quality of cell yield after harvesting and mechanical manipulation should be indicated I suggest to clarify how the tissue is tissue in combination withe dermal matrix in the study should be indicated the number of cases treated with this protocol and results.

Author Response

Point 1: In the regenerative therapy the amount of SVF cells is very important in evaluating the effectiveness of the treatment.  In this study the quantity and quality of cell yield after harvesting and mechanical manipulation should be indicated I suggest to clarify how the tissue is tissue in combination withe dermal matrix in the study should be indicated the number of cases treated with this protocol and results.

Response 1: We totally agree and thanks for the important input. Yes, SVF cells play one of the most important roles clinically, especially if tissue engineering protocols will be used for patients. We, through this ex-vivo experimental note shed the light on efficiency and safety of possible use for such products. Nevertheless, we should clearly state that further analytical research should also be initiated in order to identify the best quantity of cells needed for every square meter before transplantation.

We also clearly indicated that this paper is describing ex vivo experimental analysis which does not represent an original work or case series for the intention of treating patients. The authors clearly describe a novel approach showing promising results that can be taken further into transitional or clinical settings for more improvement in terms of tissue repair and regeneration.